# SELWAK: A Secure and Efficient Lightweight and Anonymous Authentication and Key Establishment Scheme for IoT Based Vehicular Ad hoc Networks

**DOI:** 10.3390/s22114019

**Published:** 2022-05-26

**Authors:** Sagheer Ahmed Jan, Noor Ul Amin, Junaid Shuja, Assad Abbas, Mohammed Maray, Mazhar Ali

**Affiliations:** 1Department of Computer Science and Information Technology, Hazara University, Mansehra 21300, Pakistan; saghir30232@gmail.com (S.A.J.); namin@hu.edu.pk (N.U.A.); 2Department of Computer Science, Abbottabad Campus, COMSATS University Islamabad, Abbottabad 22060, Pakistan; junaidshuja@cuiatd.edu.pk; 3Department of Computer Science, Islamabad Campus, COMSATS University Islamabad, Islamabad 44000, Pakistan; assadabbas@comsats.edu.pk; 4College of Computer Science and Information Systems, King Khalid University, Abha 62529, Saudi Arabia; mmarey@kku.edu.sa

**Keywords:** authentication, internet of things, vehicular and wireless technologies, privacy, computational efficiency

## Abstract

In recent decades, Vehicular Ad Hoc Networks (VANET) have emerged as a promising field that provides real-time communication between vehicles for comfortable driving and human safety. However, the Internet of Vehicles (IoV) platform faces some serious problems in the deployment of robust authentication mechanisms in resource-constrained environments and directly affects the efficiency of existing VANET schemes. Moreover, the security of the information becomes a critical issue over an open wireless access medium. In this paper, an efficient and secure lightweight anonymous mutual authentication and key establishment (SELWAK) for IoT-based VANETs is proposed. The proposed scheme requires two types of mutual authentication: V2V and V2R. In addition, SELWAK maintains secret keys for secure communication between Roadside Units (*RSU_s_*). The performance evaluation of SELWAK affirms that it is lightweight in terms of computational cost and communication overhead because SELWAK uses a bitwise Exclusive-OR operation and one-way hash functions. The formal and informal security analysis of SELWAK shows that it is robust against man-in-the-middle attacks, replay attacks, stolen verifier attacks, stolen OBU attacks, untraceability, impersonation attacks, and anonymity. Moreover, a formal security analysis is presented using the Real-or-Random (RoR) model.

## 1. Introduction

The past decade has witnessed colossal advancements in Information and Communication technologies (ICT) resulting in a number of concepts appearing on technological horizons. In practice, ICT has become an integral part of every field of human life. The concept of “smart and autonomous environment” is the result of emerging ICT models that can benefit human society at large. The Internet of Things enables the autonomous and smart society to connect billions of smart devices to inter- and intra-communication to achieve its goals [1,2,3]. These intelligent sensing and interconnected devices depict a tremendous capacity for replicating the physical environment into corresponding digital environments. IoT-based smart environments can assist society in a broad spectrum, such as e-health care, business, e-commerce, logistics, education, agriculture, defense, and many more.

VANETs are a crucial component of a smart and autonomous environment with an aim to deliver Intelligent Transport System [4] where vehicles communicate with each other, roadside infrastructure, and/or other network services. ITS aims to provide controlled traffic flows, co-operative traffic monitoring, collision prevention, detour route computation, and internet connectivity to moving vehicles. Therefore, VANETS became a combination of wireless ad hoc networks and IoT-based devices for the provision of services. There are three main components of ITS: (a) vehicle, (b) Trust Authority (TA), and (c) Road-Side Unit (RSU), as shown in Figure 1. Vehicular communication takes place in two ways: (a) Vehicle to Vehicle (V2V) and (b) Vehicle to RSU (V2R). Each vehicle is equipped with an onboard unit (OBU) that receives and processes traffic-related data. The OBU also transmits information related to neighboring vehicles and *RSU_s_* using Dedicated Short Range Communication (DSRC) protocols [5]. The RSU is deployed beside the road as a base station and acts as a connecting node between *OBU_s_* and the Trusted Authority (TA). The RSU performs various authentication operations. The TA’s responsibilities are to register the *OBU_s_* and *RSU_s_*, perform maintenance, and conduct the entire vehicular system.

Moving vehicles with varying accelerations make VANETs different from traditional ad hoc networks, thereby featuring specific network challenges in the case of VANETs. Resource-constrained IoT devices and the wireless nature of communication in VANETs make security a concern of prime focus [6]. Insecure communication may result in the transfer of life-critical information to an adversary. Unauthentic information may lead a passenger to a path of adversary’s choice, thus, putting life in danger [7]. Acceptance of a malicious message may cause malfunctioning of the vehicle system. Therefore, security gains prime importance in the case of VANETs, as unwanted situations may cause privacy breaches to one extent and prove to be fatal to the other.

A Secure and Efficient Lightweight Anonymous Mutual Authentication and Key establishment scheme for IoT-based vehicular ad hoc networks (SELWAK) is proposed in this paper. The proposed scheme uses a simple XOR operation and a one-way hash function, making it light in terms of resource usage. Various authentication and key establishment schemes have been discussed in the literature. Moreover, resource-constrained devices do not support traditional cryptographic operations due to low memory and computational power, and therefore demand lightweight cryptographic preemptive. Ensuring the privacy of vehicles is a challenging issue because an adversary can trace the traveling routes of vehicles and identify vehicles that may cause serious danger. To overcome privacy issues, the proposed scheme uses mask identities to ensure anonymity and privacy preservation. In addition to this, an attacker cannot relate driver’s multiple mask identities to reveal his/her real identity. The proposed scheme provides better security services in a cost-effective manner compared to existing schemes. The SELWAK consists of four phases: (i) Registration, (ii) authentication and key agreement, (iii) RSU-to-RSU key establishment, and (iv) password change.

In the registration phase, vehicles and roadside units register with the TA. The driver of the vehicle chooses various credentials and sends them to the TA in a secure way. Then, the vehicle is deployed on the VANETs. Before deployment of a vehicle in VANETs, TA sends the information to vehicle *V_i_* in a secure way, and *OBU_i_* stores that information for future use. In the RSU registration phase, the TA generates credentials for every RSU that is deployed in VANETs. The second phase consists of two sub phases, such as (i): V2V authentication key agreement phase and (ii) the V2RSU authentication key agreement phase. In each sub phase, after successful mutual authentication, a session key is established between two entities, and this key is later used for authentication purposes. In the key establishment phase of RSU-to-RSU, a session key is established between those *RSU_s_* on the basis of their preloaded credentials. For secure communication, it is necessary that the driver of the vehicle change the password periodically. There is an option available for drivers to change passwords locally without interacting with the TA. Formal security analysis of the SELWAK was done using the Real-or-Random (RoR) model. SELWAK provides better security services and effectively reduces computational cost and communication overhead, as indicated by the derived results. The following are the main contributions of this paper.

In this paper, a novel lightweight anonymous authentication and key establishment scheme for VANETs is proposed that uses one-way cryptographic hash functions and simple XOR operations.We ensure the privacy of vehicles so that an adversary cannot trace the real identity and travel routes of vehicles.SELWAK is secure against replay attacks, impersonation attacks, man-in-the-middle attacks, stolen verifier attacks, stolen OBU attacks, untraceability, and anonymity.Formal security proof of establishing a secure session key is provided using the RoR model.

The remainder of the paper is organized as follows. Section 2 discusses related work, whereas Section 3 presents systems models. In Section 4, the proposed SELWAK is described, while Section 5 presents the security analysis. In Section 6, we evaluate the performance of the proposed scheme, and Section 7 concludes the paper.

## 2. Related Work

Numerous studies exist on authentication, key establishment, and privacy preservation in VANETs. Below, we present a brief discussion of the few existing techniques. Wang et al. [8] proposed an authentication scheme for VANET using a group signature. According to the authors, when vehicles apply for group membership, membership validity is checked to determine whether the vehicle is still a member of the group. Batch verification of vehicles can also be done in the proposed scheme. The authors in [9] proposed a password based novel group key agreement protocol. Their scheme provides batter privacy services in the field of VANET. The proposed scheme uses a hash function for authentication and integrity. According to the authors, their scheme has less computational cost as well as communication overhead as compared to certificate-based public key cryptography and identity-based public key cryptography but is vulnerable to denial-of-service attacks. In a novel secure and efficient anonymous authentication scheme with a privacy preserving scheme (EAAP) [10], *RSU_s_* and *OBU_s_* use digital signatures to sign each message. The EAAP scheme uses a bilinear-pairing technique to conform to the integrity and authentication of messages. Bilinear pairing has a high computational cost compared to the cryptographic general hash function [11]. A discrete event-based threat-driven authentication scheme has been proposed to ensure secure V2I and V2V communication in [12]. To satisfy the secure communication between V2V and V2R, the proposed approach uses a session key, private key, and public key simultaneously. The authors used the Petri Nets and Veins framework for the formal analysis of their scheme. Zhang et al. [13] proposed an identity-based public key cryptographic (ID-PKC) scheme for privacy-preservation communication. The authors used bilinear pairing and ID-PKC to originate vehicular clouds and secure communication in vehicular clouds. In this scheme, a secure and anonymous dynamic vehicular cloud comes from using pseudonyms. The authors also presented a well-organized protocol that allowed cloud users to join or leave the group dynamically. Two schemes that control traffic lights intelligently using for computing were proposed in [14]. The first scheme’s security is based on Computational Diffie-Hellman puzzle hardness, and the second is based on the hash collision puzzle. After a fixed interval of time, the traffic lights generate the puzzle and verify it. For VANETs, a decentralization mutual authentication and key agreement scheme were proposed in [15]. The vehicles communicate in the cluster’s fashion and use the hash function and XOR operation. There are three types of authentication taking place: vehicles-to-cluster heads, between cluster heads and cluster heads, and roadside units. This scheme does not deliberate batch verification and privacy preservation of the signatures of multiple messages. Ibrahim et al. [16] proposed two schemes, epidemic-based and topology-based, in which RSU switches its authentication service to the nearest vehicle for the betterment of the authentication service. The topology-based scheme depends upon network analysis and computing node degree, but the scheme based on the epidemic level did not depend on network analysis. The authors have compared both schemes and show that topology-based schemes have better performance but more security threats than epidemic-based schemes. An authentication scheme with privacy preservation property based on identity was proposed in [17]. To reduce communication overhead, a registration list is used instead of the revocation list. The security features of VANET were not affected by malicious vehicles. Moreover, their scheme did not use bilinear pairing operations, which takes more execution time, thus dramatically reducing computation and communication costs. Gope et al. [18] proposed an efficient authentication scheme based on RFID with privacy features. This scheme uses a distributed IoT infrastructure for secure localization servers to facilitate smart city environments. The backend server has a full command to recognize RFID tags without any trouble. However, the problem with this scheme is that the managing server is so powerful that it can know the entire communication of RFID tags. The security of the scheme depends on the backend server. If the backend server has a strong security mechanism, then the attacker cannot get security credentials, but if backend server security is compromised, then the attacker can easily get secret information and execute a forgery attack. Second, the RFID tags did not have any physical security. A signature based on an identity scheme for authentication of V2V communication has been proposed in [19]. This scheme is based on elliptic curve cryptography. The advantage of batch signature verification is that it can authenticate a large number of vehicles at a time. This scheme uses an RoR model for security proof. According to the authors, their scheme reduces the execution time and communication burden compared to other schemes. Cui et al. [20] proposed an authentication scheme that preserves the privacy property in the field of VANET. This scheme uses ECC and identity-based signatures for both V2I and V2V communication. The authors used the binary search method and the cuckoo filter method to improve the success rate of batch signature verification. Xie et al. [21] proposed a robust and secure conditional privacy-preserving scheme using identity-based authentication. The reliability and integrity of the messages are ensured using identity-based signatures for V2V communication and V2I communication. The results of this scheme show that it has a high computational cost and communication overhead. A conditional-based privacy and authentication scheme was proposed in [22]. The prevention from side channel attacks is gained by storing sensitive data on the TPD of OBU and updating it periodically. The formal security analysis of their scheme has been shown using BAN-logic. Their approach is based on a one-way hash function and ECC; therefore, according to the authors, their scheme is efficient in terms of cost compared to existing schemes [23,24,25,26]. To ensure secure communication in VANET, an authentication scheme based on ECC that satisfies privacy preservation was proposed in [27]. In this scheme, the authors combined RSU- and TPD-based schemes to handle privacy and security issues in VANET. All the system’s public credentials and keys are preloaded in the TPD of RSU. Their scheme worked in four phases: initialization phase, mutual authentication, signing, and verification phases. Jie et al. [28] presented a chaos mapping-based full session key agreement scheme. This scheme worked in two phases. In the first phase, group key agreement was made between the cluster head and the fog server. In the second phase, a group key agreement is made among vehicle nodes. A secure and robust authentication and privacy scheme has been introduced for vehicular communication [24]. The trusted authority preloads the already computed private key in the vehicle’s TPD via a secure medium. Jalawai et al. [27] presented an authentication mechanism using elliptic curve cryptography, which satisfied conditional privacy preservation. They addressed some security and privacy concerns based on the combined usage of TPD-based schemes with RSU-based schemes. The system’s key and all the initial public parameters are preloaded in the TPD of RSU. There are some issues with privacy and security, and some attacks are also possible. Vijayakumar et al. [29] proposed an authentication and key distribution scheme for VANET. According to the authors, their scheme is efficient in terms of both computation cost and communication overhead. In addition, the vehicles that come in the orbit of RSU securely distribute the group key among the vehicles. The RSU uses the group key to send the message related to the location among the neighboring vehicles via a secure channel. Vijayakumar et al. [30] proposed a novel batch authentication and key exchange protocol based on 6G technology for VANET. In addition, their scheme reduces the load on the RSU in congested areas. An elliptic curve-based intelligent conditional privacy-preserving technique for VANET has been proposed in [31]. The authors claimed that this scheme is secure, efficient, and can easily deploy. A cuckoo filter-based authentication scheme that improved timed efficient stream loss tolerance for VANETs was proposed in [32]. The authentication information of vehicles that came under the communication range of the RSU can be saved by a cuckoo filter. This scheme provides robust, anonymous authentication and reduces costs. To provide safety in VANET, an efficient anonymous mutual authentication approach with privacy is proposed in [32]. In their scheme, the trusted authority preloaded a group of pseudonym identities and a group of private keys to each vehicle, which may cause problems for managing huge certificates, which will increase the burden for management of certificates for TA due to the limited storage capacity of the vehicle. Ren et al. [33] proposed a blockchain-based, certificateless public key signature scheme for VANET. Their scheme provides support for batch verification of signatures, and blockchains are used to protect the privacy of vehicles. Moreover, this scheme also realized the traceability property. An authentication approach for global mobility networks was proposed in [34]. This scheme is based on an elliptic curve cryptosystem and therefore takes much execution time to perform major cryptographic operations.

The schemes discussed in the literature have some problems. Due to the fast movement of vehicles in VANET, the performance of signature-based schemes is not optimal. OBU has limited storage capacity, computing power, and power. The signing and verification of road safety-related messages slows down due to heavy cryptographic operations. For example, bilinear pairing operations consume more time for message’s signing and verification process [30]. Therefore, it is difficult for RSU to verify a large number of vehicles in its range moving with high speed in a short period of time. This puts a heavy burden on the verification vehicle, and behind the current demand for an efficient and lightweight scheme that validates many traffic-related messages on V2V, V2RSU, and RSU2RSU connections in high traffic density areas without compromising safety. On the other hand, a group signature-based scheme requires registration of each vehicle with the TA and receives its private key via a secure channel. These time-consuming operations create hurdles for vehicles to change private keys easily. Therefore, the likelihood of an attack increases.

### Motivations

VANETs and vehicles travel at high speeds; therefore, the schemes mentioned in the literature are not optimal for such an environment. The OBU fixed in the vehicle has limited storage capacity, power supply, and computational power. Various major cryptographic operations slow down the signature generation and verification processes of road safety-related messages. For example, elliptic curve point multiplication and point addition are considered to be the most time-consuming operations in ECC-based schemes. Therefore, it is difficult to verify vehicles moving at high speeds by the RSU in a short time period in its communication range. It creates a high load on verifying entities, which is the reason it demands a secure and efficient lightweight and anonymous authentication and key establishment scheme for IoT-based vehicular ad hoc networks.

## 3. System Model

The network and thread models are presented in this section.

### 3.1. Network Model

The network model for VANET used in the SELWAK is shown in Figure 1. In this model, the entities involved are vehicles (*V_i_*), roadside units (*RSU_s_*), and TA. In the network model, three types of participation involved: V2V, V2RSU and RSU2RSU.The TA is responsible for generating identities, for example, keys, and identities for vehicles and *RSU_s_*. The information generated by TA is stored in the memory of *RSU_s_* and *OBU_s_*, which can be used for authentication purposes. In light of the proposed model, the authentication processes that are required are V2V, V2RSU and RSU2RSU.

### 3.2. Threat Model

According to this model, all entities are assumed to communicate with each other through the insecure channel. *RSU_s_* are also assumed to be semi-trusted. An attacker can easily delete, modify, or eavesdrop the transmitted message. As *RSU_s_* are considered semi-trusted, we considered that the RSU’s confidential information is stored in tamper-proof devices within *RSU_s_*. However, we considered that *OBU_s_* are not installed with tamper-proof devices. Moreover, by using a power analysis attack [22,23], an attacker can extract all the sensitive information from some stolen *OBU_s_* of the vehicles. Finally, the TA is considered a fully trusted authority.

## 4. Proposed Scheme

In this paper, a novel lightweight and anonymous authentication and key establishment scheme for IoT-based VANETs is proposed. In SELWAK, when a vehicle joins the region of another vehicle, anonymous mutual authentication between the vehicles is performed to avoid communication with malicious vehicles. To perform different types of wireless communications in VANETs, our authentication scheme can be divided into three categories: Vehicle-to-Vehicle, Vehicle-to-Roadside Unit, and Roadside Unit-to- Roadside Unit authentication. The proposed scheme works in four phases: registration phase, authentication, and key agreement phase, RSR-to RSU key establishment phase, and password change phase. Before giving a detailed description of the various phases, we briefly describe each phase in Figure 2. The definitions of the notations in our scheme are described in Table 1.

### 4.1. Registration Phase

In this phase, the registration of vehicles and roadside units is done in the following ways.

#### 4.1.1. Vehicle Registration Phase

It is necessary to register each vehicle offline with the TA for secure V2V and V2R communication. The vehicle’s registration with the TA is a one-time process; hence, for the execution of this process, a secure channel is required, e.g., in person. The steps below are used for this purpose.
The driver *Drv_i_* of vehicle *V_i_*, on his own choice, chooses a password *PWD_i_* and unique identity *Drv_id_* and two 160-bit random numbers *s_i_* and *k*. *OBU_i_* computes a masked password $MPWD_{i}=h(PWD_{i}\;||s_{i})$, transmit $(drv_{id},\;(MPWD_{i}⊕k))\;$to the TA through a secure channel.After receiving the registration request $(drv_{id},\;(MPWD_{i}⊕k)),\;$TA calculated $Mdrv_{id}=h(drv_{id}\;||a),\;E{\;}_{1}=h(Mdrv_{id}\;||\alpha )\;$using a pre-generated 160-bit secret key *α*. It further calculate $E_{2}\;=\;h(drv_{id}\left|\right|E_{1}\left|\right|TA_{id}),\;r\;=\;h(TA_{id}||\alpha ),\;r^{\prime }\;=\;h(TA_{id}||\beta ),\;A_{1}\;=\;r⊕E_{2}⊕(MPWD_{i}⊕k),\;\mathrm{and}\;A_{2}\;=\;r^{\prime }⊕E_{2}⊕(MPWD_{i}⊕k).\;$Furthermore, for every registered vehicle *V_i_*, a unique secret key $SeKV_{i}$ is also generated by TA and computes time based credential $TV_{i}=h(SeKV_{i}\;||RT_{vi}\;||\;drv_{id})$ on the basis of timestamp generated duringregistration time $RT_{v}\;\mathrm{of}\;V_{i}\;\mathrm{and}\;\mathrm{identity}\;drv_{id}\;\mathrm{of}\;\mathrm{driver}.\;$Then, TA transmit $\left(Mdrv_{id},\;TV_{i},\;TA_{id},\;E_{1},\;E_{2},A_{1},A_{2}\right)$ to through a secure channel.After receiving information $\left(Mdrv_{id},\;TV_{i},\;TA_{id},\;E_{1},\;E_{2},\;A_{1},\;A_{2}\right),\;OBU_{i}$ compute $f_{i}\;=\;h(PWD_{i}\left|\right|drv_{id})⊕s_{i}{,\;E}_{1}^{\prime }\;=E_{1}⊕h(drv_{id}\left|\right|s_{i}),\;{\mathrm{TA}}_{\mathrm{id}}^{\prime }h({drv}_{id}\left|\right|s_{i})TA_{id},\;E_{3}\;=\;h({drv}_{id}\left|\right|\;MPWD_{i}\left|\right|{TA}_{id}\left|\right|E_{1}),\;E_{4}\;=\;h(E_{3}\left|\right|E_{2}),\;Mdrv_{id}^{\prime }\;=\;Mdrv_{id}⊕h(PWD_{id}\left|\right|\;drv_{id}\left|\right|s_{i}),\;{\mathrm{TV}}_{i}^{\prime }\;=\;{TV}_{i}⊕h({PWD}_{i}\left|\right|s_{i}),\;A\;=\;A_{1}⊕k\;=\;r⊕E_{2}⊕MPWD_{i}.$

*OBU_i_* then deletes *k*$,\;Mdrv_{id},\;TV_{i},\;$*TA_id_*, *E*_1_, *A*_1_ and *A*_2_ from its memory. Finally, *OBU_i_* contains $\{Mdrv_{id}^{\prime },\;{TV}_{i}^{\prime },\;{\mathrm{TA}}_{\mathrm{id}}^{\prime },\;f_{i},\;Y,\;E_{1}^{\prime },\;E_{4},\;h(\cdot )\}.$ The pictorial representation of algorithm is given in Figure 3.

#### 4.1.2. Roadside Unit Registration Phase

Trusted authority generates 160-bit secret keys *α* and *β*, before deployment of $RSU_{s}$ in VANETs. Then trusted authority generates unique identities of $RSU_{s}$ like $RSU_{id1}$, $RSU_{id2}$ … $RSU_{idn}$ and corresponding masked identities $\gamma_{i},\gamma_{j}\;\ldots \;\gamma_{n}$ that are generated as $\gamma =h\;(RSU_{idk}||\beta ).$ The TA further generates identities for $RSU_{j}$ as $r{\;}^{\prime }=h(TA_{id}||\;\beta )$. In addition, TA generates time-based identities for each $RSU_{j}$ as $T{\mathrm{RSU}}_{j}=h\;\left({\mathrm{TA}}_{\mathrm{id}}\left|\left|{\mathrm{RTRSU}}_{j}\right|\right|\;\beta \right)$. The $RSU_{j}$ then give the information $\left\{r,\;\gamma ,TRSU_{j}\right\}.$ In our scheme γ is used for Vehicle *V_i_* to $RSU_{j}$ authentication and $TRSU_{j}$ is used for symmetric key establishment between $RSU_{s}$. The polynomial-based key distribution for *RSU2RSU* key establishment. To do this, TA first selects bivariate polynomial þ(x,y) = $\;þ\left(x,y\right)=\overset{n}{\underset{l}{\sum }}0\overset{n}{\underset{m=0}{\sum }}s^{l},m^{x^{l}y^{m}}\in GF\left(þ\right)\left[x,y\right]$ over a finite field degree n. For each $RSU_{j}$ TA computer polynomial share $þ\left(TRSU_{j},\;y\right).$ The $RSU_{j}$ is also loaded with $þ\left(TRSU_{j},\;y\right)$ in its memory.

### 4.2. Authentication and Key Establishment Phase

Initially, $Drv_{i}$ inputs a password ${\mathrm{PWD}}_{i}^{*}$ and identity *drv_id_* to $OBU_{i}$. The $OBU_{i}$ calculates $s_{i}^{*}=f_{1}\;⊕h(PWD_{i}^{*}\;||drv_{id}$), $E_{1}^{*}=E_{1}^{\prime }⊕\;h(drv_{id}||s_{i}^{*}$) $=h(Mdrv_{id}\;||\alpha )$, $MPDW_{i}^{*}$ = *h*($PWD_{i}^{*}$ ||$s_{i}^{*}$), $TA_{id}^{*}$ = $TA_{id}^{\prime }⊕$*h*($drv_{id}$ ||$s_{i}^{*}$) and $Mdrv_{id}$ = $Mdrv_{id}^{\prime }⊕\;$*h*($PWD_{i}^{*}$||$drv_{id}$||$s_{i}^{*}$). $OBU_{i}$ further computes *E_2_** = *h* ($Mdrv_{id}$||$E_{1}^{*}$||${TA}_{id}^{*}$), $r=A\;⊕\;E_{2}^{*}⊕\;MPDW_{i}^{*}$, $r\prime \;=A\prime ⊕\;E_{2}^{*}\;⊕\;MPDW_{i}^{*}$,$E_{3}^{*}$ = *h* ($drv_{id}$||$MPDW_{i}^{*}$||$TA_{id}^{*}$||$E_{1}^{*}$) and $E_{4}^{*}$ = *h* ($E_{3}^{*}$||$E_{2}^{*}$). Inputting correct credentials: password and identity by authorized users. Each vehicle also computes the same r and r’. *OBU_i_* checks the condition if $E_{4}^{*}$ = *E*_4_. If conditions hold, it implies that $drv_{i}$ is authentic users. If the condition is not satisfied, then the phase is terminated. In addition, $OBU_{i}$ also computes $TV_{i}=TV_{i}^{\prime }\;⊕\;MPDW_{i}^{*}$.

#### 4.2.1. V-To-V Authentication and Key Establishment Phase

In V2V authentication, two neighboring vehicles perform the following steps:Onboard Unit *OBU_i_* generates current timestamp *T*_1_ and chooses random nonce $N_{OBUi},\;$and computes secret key $KSr_{1}=h(r\;||T_{1}).\;$Two neighbor vehicles used *r* and $r^{\prime }$ for authentication in VANETs. An *OBU_j_* further compute $J_{1}=h(N_{OBUi}\;||\;Mdrv{\;}_{id}||\;TV{\;}_{i}\left|\right|T_{1}),\;L_{1}=KSr_{1}⊕J_{1}$ and $L_{2}=h(J{\;}_{1}{||\mathrm{TA}}_{\mathrm{id}}^{*}\;||T_{1}),\;$and sends authentication requests {*L*_1_, *L*_2_, *T*_1_} to its neighboring vehicle through a public channel.“After receiving {*L*_1_, *L*_2_, *T*_1_}, *OBU_j_* validates the timeliness of *T*_1_ by checking condition |T1− T1*| ≤ ΔT, where T1* is the time when the message is received and Δ*T* is the maximum transmission delay. If the condition holds, *OBU_j_* calculates the time-dependent secret key $KSr_{1}=h(r\;||T1)$ on the basis of *T*_1_ and previously computed *r*. It then computes $J_{1}^{\prime }=KSr_{1}⊕L_{1}=h\;(N_{OBUi}\;||\;Mdrv_{id}\;||\;TV_{i}\;||T_{1}).\;$To proceed, it then calculates $L_{3}=h\;(J_{1}^{\prime }\left|\right|TA_{id}^{*}\;||T_{1}).\;$The *OBU_i_* further checks the condition *L*_3_ = *L*_3_, if condition holds then *V_j_* authenticate *V_i_* and reject otherwise.The *OBU_j_* selects a random nonce $N_{OBUi}$ and current timestamp *T*_2_, and computes time-dependent secret key $KSr_{2}=h(N_{OBUj}\;||T_{2}),\;J_{2}=h\;(N_{OBUj}\;||\;Mdrv_{idj}\;||\;TV_{i}\;||T_{1}\;||T_{2})\;\mathrm{and}\;L{\;}_{4}=\;TV_{i}⊕J_{2}.\;$Then, the session key is computed $S_{kvv}=h(h(r||T{\;}_{1}\left|\right|T_{2})||\;J_{1}^{\prime }\;||J_{2}\;||\;TA_{id}^{*})\;\mathrm{and}\;L_{5}=h\;(S_{kvv}\;||T_{2}),$ and sends {*L*_4_, *L*_5_, *T*_2_} to *V_i_* via a public channel.On the reception of {*L*_4_, *L*_5_, *T*_2_}, *OBU_i_* also checks the validity of *T*_2_ by |T2−T2*| ≤ΔT, where $T_{2}^{*}$ I message arrival time. If the condition is fulfilled, by using received *T*_2_ and earlier computer *r* and $J_{2}^{\prime }=KSr_{2}⊕L_{4}=h(N_{OBUj}||\;Mdrv_{idj}\;||TV_{j}\;||T_{1}\;||T_{2}).$, *OBU_i_* computes $KSr_{2}=h\;(r||\;T_{2})$. The *OBU_i_* further computes the session key $S_{kvv}^{\prime }=h(h(r\;||T_{1}\;||T_{2})||\;J_{1}\;||J{\;}_{2}^{\prime }||\;TA_{id}^{*}),\;L_{6}=h(S_{kvv}^{\prime }\;||T_{2}).$ It then checks the condition *L*_6_ = *L*_5_. If the condition is satisfied, *V_i_* successfully authenticates_._ Using the current timestamp *T*_3_, the OBU computes $L_{7}=h(S_{kvv}^{\prime }\;||T_{3}),\;$and finally sends a response message {*L*_7_, *T*_3_} to *V_j_* via a public channel.On the reception of {*L*_7_, *T*_3_}, *OBU_j_* checks the correctness of *T*_3_ by checking condition |T3−T3*| ≤ ΔT, where T3* is reaching time. Then, it computes $L_{8}=h(S{\;}_{kvv}\left|\right|T_{3})\;$and checks whether $L{\;}_{8}=L_{7}.\;$If the condition is satisfied, the session key computed by *OBU_i_* is correct, and it guarantees that both *V_i_* and the session key are established by *V_j_* in this way $S_{kvv}\;(=S_{kvv}^{\prime })\;\mathrm{to}\;$start mutual communication. The pictorial representation of algorithm is given in Figure 4.

#### 4.2.2. V-to-RSU Authentication and Key Establishment Phase

In this phase, vehicle $V_{i}$ and neighbor roadside unit *RSU_j_* perform the following steps for authentication and key establishment:An *OBU_i_* chooses a timestamp *T*_1_ and random nonce $NV_{i}$ and calculates the time-dependent key $SK_{r}^{\prime }=h(r^{\prime }\;||T1)\;$on the basis of previously calculated *r*. It further computes $J_{1}=h(NV_{i}\;||\;Mdrv_{id}\;||TV_{i}\;||T_{1}),\;L{\;}_{1}=SK_{r1}^{\prime }⊕J_{1}\mathrm{and}\;L_{2}=h(J_{1}\left|\right|TA{\;}_{id}^{*}\left|\right|T_{1})\;$and sends {*L*_1_, *L*_2_, *T*_1_} as an authentication message to its nearby *RSU_j_* through a public channel.After receiving {*L*_1_, *L*_2_, *T*_1_} *RSU_j_* validate *T*_1_. If it validates the timestamp, then *RSU_j_* calculates the time-dependent key $SK_{r1}^{\prime }=h(r^{\prime }\left|\right|T_{1})\;\mathrm{on}\;\mathrm{the}\;\mathrm{basis}\;\mathrm{of}\;T_{1}.\;\mathrm{It}\;\mathrm{then}\;\mathrm{computes}J_{1}^{\prime }=SK_{r}^{\prime }⊕L_{1}=h(NV_{i}\;||\;Mdrv_{id}\;||TV_{i}\;||T_{1})\;\mathrm{and}\;L_{3}=h(J_{1}^{\prime }\;||\;TA{\;}_{id}^{*}\left|\right|T_{1}).\;$If *L*_3_ = *L*_2_ holds the *RSU_j_* authenticate *V_i_* and reject otherwise.The *RSU_j_* then chooses the current timestamp *T*_2_ and random nonce *N_RSU_* to calculate another time-dependent key $KS_{r}=h(r^{\prime }\left|\right|T_{2}),J_{2}=h(N_{RSUj}\;||\;\gamma \;||T_{1}\;||T_{2})\;\mathrm{and}\;L_{4}=KS_{r}⊕J_{2}.\;$It further calculates the session key $S_{kVR}=h(h(r^{\prime }\left|\right|T_{1}\;||T_{2})||\;J_{1}^{\prime }||\;J_{2}||\;TA{\;}_{id}^{*})$ and $L_{5}=h(S_{kVR}\;||T_{2}),$ and sends message {*L*_4_, *L*_5_, *T*_2_} to *V_i_* through an open channel. The pictorial representation of algorithm is given in Figure 5.

### 4.3. Key Establishment Phase between RSU_s_

Two neighbor Roadside Units, namely *RSU_u_* and *RSU_v_* established pairwise key using the following steps.

The random nonce $N_{RSU}{}_{u}$ is generated by *RSU_u_* and sends $\{TRSU_{u},\;N_{RSU}{}_{u}\}\;$to *RSU_v_*.Upon receiving “$\{TRSU_{u},\;N_{RSU}{}_{u}\},\;$*RSU_u_* calculates symmetric key shared with *RSU_u_* as $S_{kRR}=þ\;(TRSU_{v},TRSU_{u})$ by pre-loaded polynomial share *þ* (*TRS_v_*, *y*) and $S_{KV}=h(S_{kRR}\;||N_{RSU}{}_{u}).\;$The *RSU_v_* then sends the message $\{TRSU_{u},S_{KV}{\}\;\mathrm{to}\;\mathrm{RSU}}_{u}.$Finally, on reception of ${\{\mathrm{TRSU}}_{u}{,\;S}_{\mathrm{KV}}\},\;$*RSU_u_* calculate the symmetric key and share with *RSU_u_*$\mathrm{as}\;S_{kRR}^{\prime }=þ\;(TRSU_{u},TRSU_{v})\;(=S_{kRR})\;$by pre-loaded polynomial share *þ* (*TRSU_u_*, *y*) and $S_{KV}^{\prime }=h(S_{kRR}^{\prime }\;||\;N_{RSU}{}_{u})$ on the basis of its own already generated random nonce $N_{RSU}{}_{u}.\;$In addition to this, *RSU_u_* proves if $S_{KV}^{\prime }=S_{KV}.\;\mathrm{If}\;\mathrm{the}\;\mathrm{condition}\;\mathrm{is}\;\mathrm{satisfied},\;$it showed that both *RSU_u_* and *RSU_v_* used valid symmetric keys for their onward communication.After receiving {*L*_4_, *L*_5_, *T*_2_}, *OBU_i_* also validates *T*_2_. If it is valid, then *OBU_i_* calculate time-dependent key $SK_{r2}^{\prime }=h\;(r^{\prime }\left|\right|T_{2})\;\mathrm{on}\;$the basis of *T*_2_ and $J_{2}^{\prime \;}=SK_{r}^{\prime }⊕L_{4}=h(N_{RSUj}\;||\;\gamma \;||T_{1}\;||T_{2}).$ It further calculates a session key $S_{kVR}^{\prime }=h(h(N_{RSUj}\;||\;\gamma \;||T_{1}\;||T_{2})||\;J_{1}||\;J_{2}^{\prime }\;||TA{\;}_{id}^{*})\;$and $L_{6}=h(S_{kVR}^{\prime }\;||T_{2}).\;\mathrm{If}\;\mathrm{condition}\;L_{6}=L_{5}$ is satisfied then *V_i_* successfully authenticate *RSU_j_*. The *OBU_i_* again generates the current timestamp *T*_3_ to calculates $L_{7}=h(S_{kVR}^{\prime }\;||T_{3})$ and sends {*L*_7_, *T*_3_} to *RSU_j_* through an open channel.Upon receiving a message {*L*_7_, *T*_3_}, *RSU_j_* Validates *T*_3_. If it is valid, then *RSU_j_* calculates $L_{8}=h(S_{kVR}\;||T_{3})$ and checks whether *L*_8_ = *L*_7_. If the condition is satisfied, then the session key computed by *OBU_i_* is correct.

### 4.4. Password Update Phase

In SELWAK, after the registration phase, the Vehicle’s $OBU_{i}$ can update password without using a verification table. The legal user changes the password periodically to improve the security of the system. The following steps are used:*Drv_i_* provides provides an identity *drv_id_* and an old password $PWD_{i}^{old}.\;\mathrm{The}\;OBU_{i}\;\mathrm{then}\;\mathrm{computes}\;s_{i}^{*}\;=\;f_{i}⊕h(PWD_{i}^{old}\;||drv_{id}),\;E_{1}^{*}\;=\;E_{1}^{\text{'}}⊕h(drv_{id}\;||\;s_{i}^{*}),\;MPWD_{i}^{old}\;=\;h(PWD_{i}^{old}\;||\;s_{i}^{*}),\;TA_{id}^{*}\;=\;TA_{id}^{\text{'}}⊕h(drv_{id}||\;s_{i}^{*}),Mdrv_{id}^{*}\;=\;Mdrv_{id}^{\text{'}}\;⊕\;h(PWD_{i}^{old}\;||drv_{id}\;||s_{i}^{*}),\;E_{2}^{*}\;=\;h(Mdrv_{id}^{*}\;||E_{1}^{*}||\;TA_{id}^{*}),\;E_{3}^{old}\;=\;h(drv_{id}||\;MPWD_{i}^{old}\;||\;TA_{id}^{*}\;||\;E_{1}^{*})\;\mathrm{and}\;E_{4}^{old}\;=\;h(E_{3}^{old}\;||\;E_{2}^{*}).$${\mathrm{OBU}}_{i}\;\mathrm{checks}\;\mathrm{if}\;E_{4}^{old}=E_{4}.\;$ If the condition is not satisfied, the password updating process is stopped. Else, $Drv_{i}\;\mathrm{is}\;a\;\mathrm{authentic}$ user and allowed the *OBU_i_* to update the password.The driver *Drv_i_* is requested to give a new password $PWD_{i}^{new}.\;$Then, it computes
$Mdrv_{id}^{**}\;=\;Mdrv_{id}^{*}⊕h(PWD_{i}^{new}\left|\right|drv_{id}\left|\right|s_{i}^{*}),\;TV_{i}^{*}\;=\;TV_{i}^{\prime }⊕MPWD_{i}^{old},\;TV_{i}^{**}\;=\;\;TV_{i}^{*}⊕h(TV_{i}^{*}⊕s_{i}^{*}),\;f_{i}^{new}\;=\;h(PWD_{i}^{new}\left|\right|drv_{id}⊕s_{i}^{*})),\;MPWD_{i}^{new}\;=\;h(PWD_{i}^{new}\left|\right|s_{i}^{*}),\;\;E_{3}^{new}\;=\;h(drv_{id}\left|\right|MPWD_{i}^{new}\left|\right|TA_{id}^{*}\left|\right|E_{1}^{*}),\;E_{4}^{new}\;=\;h(E_{3}\left|\right|E_{2}^{*}),\;\;A^{*}\;=\;A⊕(MPWD_{i}^{old}⊕PWD_{i}^{new})\;=\;r⊕E_{2}⊕PWD_{i}^{new}\;\mathrm{and}\;A**\;=\;A\prime ⊕(PWD_{i}^{old}⊕PWD_{i}^{new})\;=\;\;r^{\prime }⊕E_{2}⊕PWD_{i}^{new}.$Finally, *OBU_i_* replaces $PWD_{i}^{\prime },\;TV_{i}^{\prime },\;f_{i},\;A,\;A^{\prime }\;\mathrm{and}\;E_{4}$ with $drv_{id}^{**},\;TV_{i}^{**},\;f_{i}^{new},\;A^{*},\;A^{**}\;\mathrm{and}\;E_{4}^{new}$ in its memory. Therefore, *OBU_i_* contains the message $\{Mdrv_{id}^{**},\;TV_{i}^{**},\;TA_{id}^{\prime },\;f_{i}^{new},\;A^{*},\;A^{**},A_{1}^{\prime },\;E_{4}^{new},\;h(\cdot )\}$ after the password update. The pictorial representation of algorithm is given in Figure 6.

## 5. Security Analysis

The RoR model [21] was used for the formal security analysis of SELWAK. We also show that our scheme is secure against well-known attacks.

### 5.1. Formal Security Analysis

Formal security analysis of SELWAK is presented using the Real-or-Random (RoR) model. The security of the session key is shown using the RoR model for the proposed scheme. There are two main participants in our scheme: Vehicle *V_i_* and Roadside Unit $RSU_{j}$. The RoR [35] has the following components.

#### 5.1.1. Participants

Let ${ῃ}_{vi}^{t}$ and ${ῃ}_{RSUj}^{u}$ be the instance t and u of the *V_i_* and $RSU_{j}$, and called as oracles.

#### 5.1.2. Accepted State

The ${ῃ}^{t}$ is an instance that is called an accepted state. Upon reception of the last message, it changes into an accepted state. The ${ῃ}^{t}$ concatenate the entire sent and received messages in proper order and for the current session form a session identification of ${ῃ}^{t}$.

#### 5.1.3. Partnering

Two of the instances ${ῃ}^{t1}$ and ${ῃ}^{t2}$ are called the partners of each other if they fulfill the following conditions.

Both of ${ῃ}^{t1}$ and ${ῃ}^{t2}$ are in valid accepted states.Both of ${ῃ}^{t1}$ and ${ῃ}^{t2}$ mutual authenticate and share identical session identification.Both of ${ῃ}^{t1}$ and ${ῃ}^{t2}$ are mutual partners [36].

#### 5.1.4. Freshness

If attacker A cannot apply the key generated for a particular session of two nodes on the bases reveal query then ${ῃ}_{vi}^{t}$ and ${ῃ}_{RSUj}^{u}$ are called fresh.

#### 5.1.5. Adversary

Adversary A has full control over the communication between the partners and has the ability to alter the message. Adversary has the following access to queries:EX (${ῃ}_{vi}^{t}$, ${ῃ}_{RSUj}^{u}$): An adversary executes this query to obtain a message that is exchanged between two original partners. This is called an eavesdropping attack.RL (${ῃ}^{t}$): An adversary using this query gets the current session key generated by ${ῃ}^{t}$.SN (${ῃ}^{t}$, message): By executing this query, an adversary sends a message to the participant and receives the message. This is called an active attack.OBU (${ῃ}_{vi}^{t}$): An adversary executes this query to extract stored information in OBU. This is called a stolen attack.Test (${ῃ}^{t}$):It models the semantic security ofa session key. After starting the experiment, coin *c* is flipped, and only the adversary can know the output. This is helpful for determining the output of a test query.

#### 5.1.6. Session Key’s Semantic Security

The main task of an attacker is to differentiate the real session key from the random session key of an instance in the RoR model. An adversary has several test queries to either ${ῃ}_{vi}^{t}$ and ${ῃ}_{RSUj}^{u}$. The random bit *c* and the output of the test query should be consistent. When an experiment is over, an adversary outputs a guessed bit c′ and wins the game if c′=c. Suppose Win is an event in which an adversary can win a game. The advantage of Adversary is that it breaks the semantic security of the proposed authentic key exchange schemes. Authentic key exchange is defined by $ad_{TA}^{AKE}=\left|2pr\left[Win\right]-1\right|$. TA is secure if $ad_{TA}^{AKE}\leq \theta$ for a sufficient smart real number θ > 0.

#### 5.1.7. Random Oracle

All the participants, including the adversary, will have to access a one-way hash function, which is called the random oracle model [36].The security proof of Theorem 1 presented in [20] is the same. The breaking of the semantic security of the session key for V2V and V2R is proved in Theorem 1 [37].

**Theorem** **1.***In the RoR model, intruder A runs in polynomial time t against the SELWAK. Let Q_h_, |Hash|, Dec, |Dec| and Q_SN_ be a number of the H queries, the range space of h(·), distributed password dictionary, size of dictionary, and number of sent queries. An adversary’s advantage* $ad_{TA}^{AKE}$*break the semantic security of the session key between OBU and RSU in the proposed scheme is defined as*(1)adTAAKE≤Qh2/|Hash|+2.QSN|Dec|.

**Proof.** As in the Chang and Le scheme [36], here the sequences of the four games says G_i_ = (0,1,2,3). *Win_i_* is an event where an adversary can successfully guess a bit c in game G_i_. Below is a detailed description of these games. □

**Game G_0_:** In the random oracle model, it is considered a real attack of the adversary on the proposed scheme. An adversary first guess bit c at the start of the game. By definition, we have
(2)$$ad_{RSU}^{AKE}=\left|2prb\left[Win_{0}\right]-1\right|$$

**Game G_1_:** In this game, an eavesdropping attack of an adversary is simulated by executing an EX (${ῃ}_{vi}^{t}$, ${ῃ}_{RSUj}^{u}$) query. At the end of the game, the adversary makes a test query. An adversary will have to know whether the test query’s output is the real session key of the vehicle and RSU or a random number. We get
(3)$$Prb\left[Win_{0}\right]=Prb\left[Win_{1}\right]$$

**Game G_2_:** In this game, an active attack on an adversary is simulated. An adversary tries to cheat the participants to receive the altered message. To verify the collision in the hash output, an adversary is allowed to query several oracles. When the birthday paradox is applied, we have
(4)$$\left|Prb\left[Win_{1}\right]-Prb\left[Win_{2}\right]\right|\leq Q_{h}^{2}/2\left|Hash\right|$$

**Game G_3_:** In this game, the Corrupt OBU query is simulated. An adversary extracts the information stored in *OBU_i_*. It is difficult to calculate the correct password. If the system only allows a specific password as an input, we can get
(5)|Prb[Win2]−Prb[Win3]|≤QSN|Dec|

An adversary can simulate all the games except that an adversary needs to guess c to win the game after the test query to oracle; we get $Prb\left[Win_{3}\right]=1/2$ from Equation (1), we have
(6)$$\left(1/2\right)ad_{RSU}^{AKE}=\left|prb\left[Win_{0}\right]-1/2\right|.$$

With the help of triangular inequality, we have |Prb[Win1]−Prb[Win3]|≤|Prb[Win1]−Prb[Win2]|+|Prb[Win2]−Prb[Win3]|≤Qh2/2|Hash|+QSN|Dec|. As a result, Equations (2) and (6) become
(7)|prb[Win0]−12|≤Qh2/2|Hash|+QSN|Dec|.

Finally, from Equations (6) and (7). we get adTAAKE≤Qh2/|Hash|+2.QSN|Dec|.

### 5.2. Informal Security Analysis

In this section, the proposed scheme’s resilience against some well-known attacks is discussed, and the security features of the proposed scheme are also compared with existing schemes.

*Replay Attack*: In the V2V and V2RSU authentication processes, the corresponding messages MSG_1_ = (*L*_1_, *L*_2_, *T*_1_) and MSG_2_ = (*L*_7_, *T*_3_) have timestamps *T*_1_ and *T*_3_. If an attacker wants to reply to the message with delay, then the timestamp attached to the message will fail. Therefore, our scheme is robust against reply attacks.*Impersonation Attack*: During the V2V authentication an attacker can impersonate the vehicle; to do so, an attacker must create an authentic message MSG_1_ = (*L*_1_, *L*_2_, *T*_1_). For creating MSG_1_ an attacker requires secret *r*. An attacker cannot calculate message MSG_1_ even if he/she generates his/her own timestamp and random none as secret *r*, Mdrv_id_, T*V_i_* and TA_id_.*Man-in-the-middle Attack*: In the proposed scheme, two messages, namely MSG_1_ = (*L*_1_, *L*_2_, *T*_1_) and MSG_2_ = (*L*_7_, *T*_3_) are required for V2V authentication. If an attacker wants to modify the message, then he/she first generates a current timestamp and random nonce. An attacker cannot calculate KS_r1A_ = *h*(*r*||*T*_1A_ as he/she did not have a secret key. Thus, an attacker cannot modify messages.*Stolen Verifier Attack*: The information (${\mathrm{Mdrv}}_{id}^{\prime }$, ${\mathrm{Mdrv}}_{id}^{\prime }$, ${TV}_{i}^{\prime }$, ${TA}_{id}^{\prime }$, *f_i_*, *Y*, $E_{1}^{\prime }$, *E*_4_, *h*(·)) is stored in *OBU_i_* of the vehicle. We assume that an attacker can steal stored information from *OBU_i_*. However, the one-way hash function protects the secrets *PWD_i_*, *r*, *r*’, *TA_id_*, *drv_id_*. An attacker cannot guess the secrets PWD_i_, *r*, *r*′, *TA_id_*, *drv_id_* correctly due to the collision resistance property of a one-way hash function.*Stolen OBU Attack*: Suppose that an attacker has stolen the *OBU_i_* of the vehicle. An attacker can extract the stored information (${\mathrm{Mdrv}}_{\mathrm{id}}^{\prime }$, ${\mathrm{Mdrv}}_{\mathrm{id}}^{\prime }$, ${\mathrm{TV}}_{i}^{\prime }$, ${\mathrm{TA}}_{\mathrm{id}}^{\prime }$, *f_i_*, *Y*, $E_{1}^{\prime }$, *E*_4_, *h*(·)) from *OBU_i_*. It is difficult for an attacker to drive *drv_id_* from *Mdrv_id_* without having the secret *α*.*Untraceability*: In the V2V and V2RSU authentication phases of the proposed scheme, two messages are followed: MSG_1_ = (*L*_1_, *L*_2_, *T*_1_) and MSG_2_ = (*L*_7_, *T*_3_). All messages are distinct in each session, and the attacker cannot trace the RSU or vehicle.*Anonymity*: In the proposed scheme, the messages for V2V and V2RSU authentication do not involve the identities of the RSU and the user. Therefore, it is infeasible for an attacker to drive the real identities of the RSU and the user. Hence, the proposed scheme satisfies the anonymity property.*Insider Attack*: SELWAk is robust against insider attacks. The neighboring vehicles cannot get unauthorized access to the sensitive information of a particular vehicle by stealing its credentials.

## 6. Performance Analysis

In this section, the performance of the proposed scheme and the existing schemes are analyzed. The proposed scheme is implemented with the following specifications: 2.66 GHz Intel(R) Core TM 2 Quad processor with 4 GB of memory using Windows 10. We compared SELWAK with some existing schemes based on computational costs, as well as communication costs. The performance result shows that our scheme is efficient in terms of computational cost and communication overhead compared to existing schemes.

### 6.1. Computation Overhead

The notations *T*_pm_-ECC, *T*_pa_-ECC, and *T*_h_ used in Table 2 represent Elliptic Curve Cryptographic points multiplication, Elliptic Curve Cryptographic points addition, and one-way hash function, respectively. As bitwise XOR operations take negligible time, we have not considered them for performance evaluation.

We have considered the values 0.6718 ms, 0.0031 ms, and 0.001 ms for various cryptographic operations like *T*_pm_-ECC, *T*_pa_-ECC, and *T*_h_ from existing experimental values [5,19,27]. The computational costs of SELWAK and some existing schemes are compared in Table 2. The schemes to which we compare our work include those of Zhong et al. [17], Ali et al. [19], Cui et al. [20], Xie et al. [21], Li et al. [24], Al-shareeda et al. [27], and Jalawai et al. [32]. An authentication scheme with privacy preservation property based on identity was proposed in [17]. To reduce communication overhead, a registration list is used instead of a revocation list. The security features of VANET were not affected by malicious vehicles. Moreover, their scheme did not use bilinear pairing operations, which takes more execution time. An elliptic curve cryptography-based and identity-based signature with a conditional privacy-preserving authentication scheme and general one-way hash functions for V2V communication is proposed in [19]. Cui et al. [20] presented a secure authentication approach with privacy properties for VANET. This scheme uses ECC and identity-based signatures for both V2I and V2V communication. The authors used the binary search method and the cuckoo filter method to improve the success rate of batch signature verification. Xieet al. [21] proposed a robust and secure conditional privacy-preserving scheme using identity-based authentication. The reliability and integrity of the messages are ensured using identity-based signatures for V2V and V2I communication. Performance analysis shows that this scheme has a high computational cost and communication overhead. To ensure secure communication in VANET, an authentication scheme based on ECC that satisfies privacy preservation is proposed in [27]. An efficient, provably-secure and anonymous conditional privacy-preserving authentication scheme for vehicular ad hoc networks has been proposed in [32]. Similarly, an authentication approach for global mobility networks was proposed in [38]. This scheme is based on an elliptic curve crypto system and therefore takes much execution time to perform major cryptographic operations.

The total computational cost for SELWAK is 16*T*_h_ + 11*T*_XOR_, which is less than that of all compared schemes. The performance result shows that our scheme is efficient in terms of computational cost and communication overhead compared to existing schemes.

### 6.2. Communication Overhead

In this section, we have compared our scheme with [17,19,20,21,24,27,32], schemes. The authentication message of [17] is {T, m, σ}. Thus, the size of the authentication message is 160 × 2 + 4 = 352 bits. In [19] the size of the authentication message is 2 × 40 + 2 × 20 + 4 + 160 = 1152 bits. In [20] the size of message authentication is 40 + 2 × 20 + 4 + 160 + 256 = 1084 bits. The communication cost analysis shows that the corresponding authentication message of [21] scheme is [*T*_i_, δ]. Thus, the size of the message is 320 × 2 + 100 × 2 + 32 = 992 bits. In our scheme, the authentication and key establishment phase require two messages MSG_1_ = (*L*_1_, *L*_2_, *T*_1_) and MSG_2_ = (*L*_7_, *T*_3_) and need (160 + 160 + 32) = 352 bits and (160 + 32) = 192 bits. Thus, the total computational cost for V2V and V2RSU authentication phases is equal to (352 + 192) = 544 bits. The communication overhead of various schemes have been shown in Table 3.

As shown in Figure 7, the execution time taken by our proposed scheme is much less than that of the other four schemes. The proposed scheme is also efficient, even in the worst case, compared to other schemes.

In Figure 8, we show total extra bits sent with the original message during vehicle communication for various schemes.

## 7. Conclusions

We proposed a novel SELWAK scheme for VANETs. Our scheme is efficient in terms of computational cost and communication overhead due to the one-way hash function and bitwise XOR operations. The SELWAK has extra features, such as mutual authentication and Vehicles and roadside unit anonymity properties. The proposed scheme is robust against driver impersonation attacks, OBU impersonation attacks, OBU capture attacks, RSU impersonation attacks, anonymity, and untraceability, perfect forward and backward secrecy, eavesdropping attacks, and insider attacks. The formal analysis of the proposed scheme was conducted using the RoR model. Therefore, the proposed scheme works efficiently for intelligent transportation systems.

In future work, anonymous mutual authentication will be carried out using BAN Logic and some simulation platforms, such as NS2, SUMO, and OMNET++, to simulate VANETs.

## Figures and Tables

**Figure 1 sensors-22-04019-f001:**
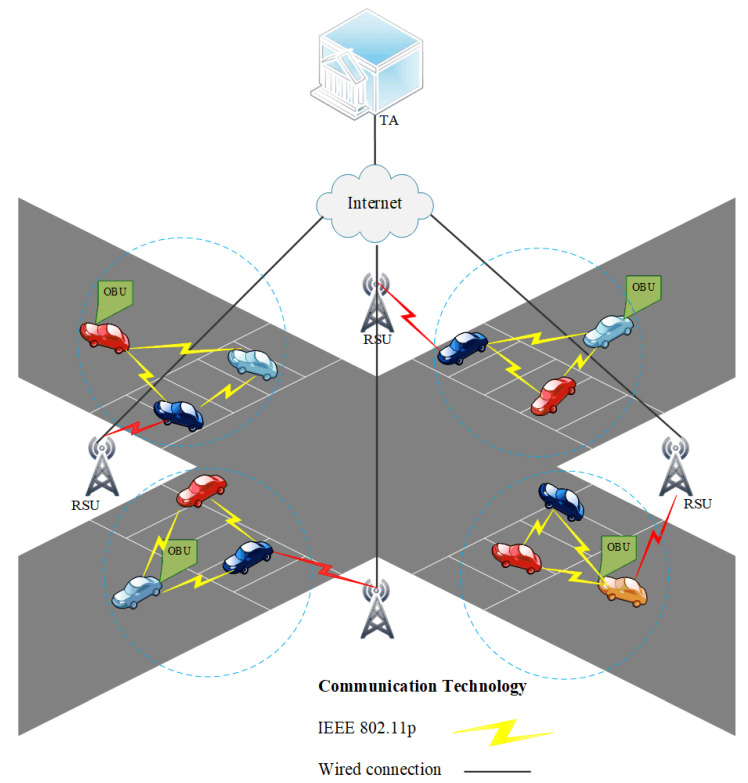
A Typical VANET Scenario.

**Figure 2 sensors-22-04019-f002:**
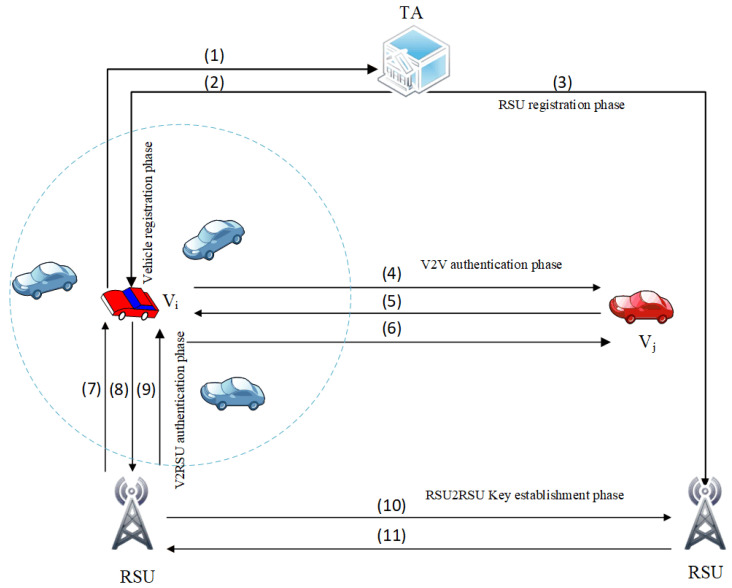
The phases involved in the proposed scheme. Vehicle Registration Phase (1). Registration request message (2). The registration response message. RSU Registration Phase (3). RSU’scredentials generated by TA. V2V Authentication and Key Establishment Phase (4). Authentication request message (5). Authentication reply message (6). Acknowledge message. V2RSU Authentication and Key Establishment Phase (7). Send a request message for Authentication (8). Authentication reply message (9). Acknowledgement message RSU2RSU Key Establishment phase (10). Send a request message for Key establishment (11). Key establishment response message.

**Figure 3 sensors-22-04019-f003:**
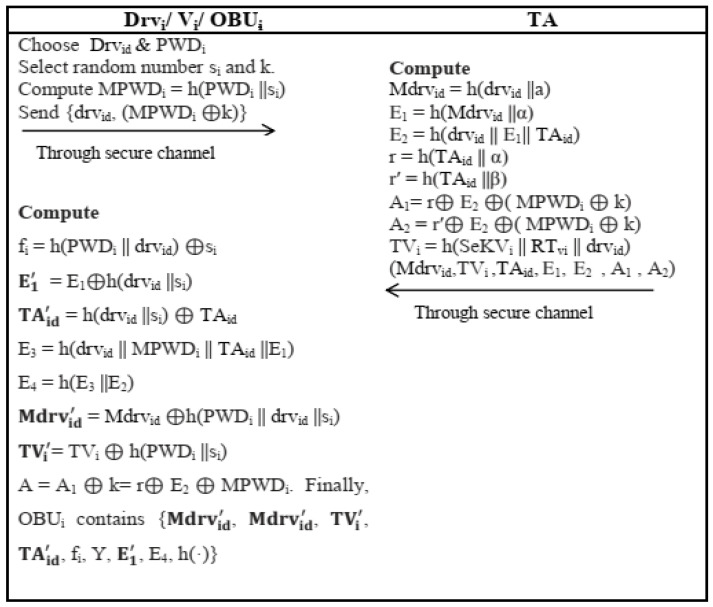
Vehicle Registration Phase.

**Figure 4 sensors-22-04019-f004:**
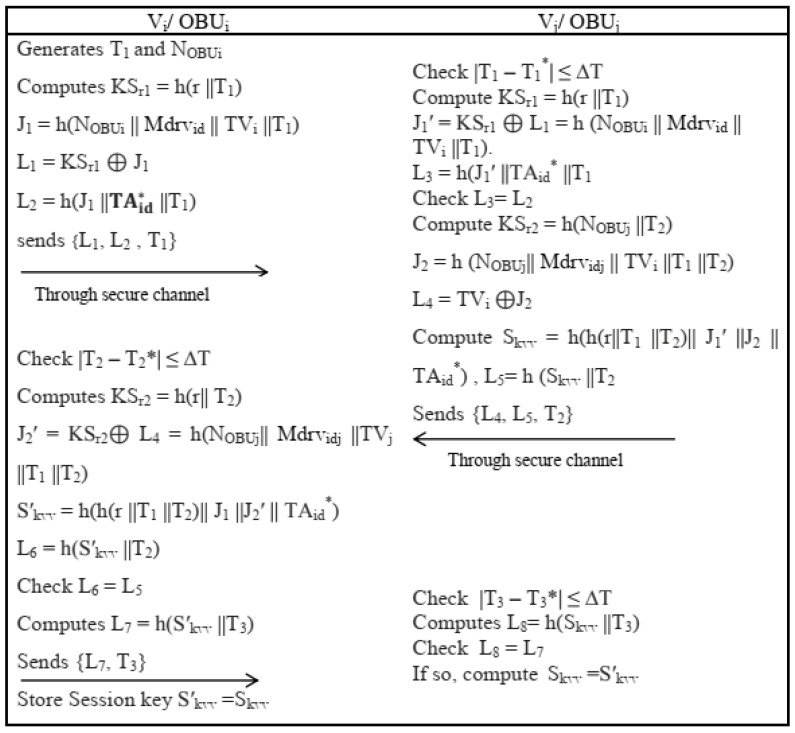
V2V Authentication and Key Establishment Phase.

**Figure 5 sensors-22-04019-f005:**
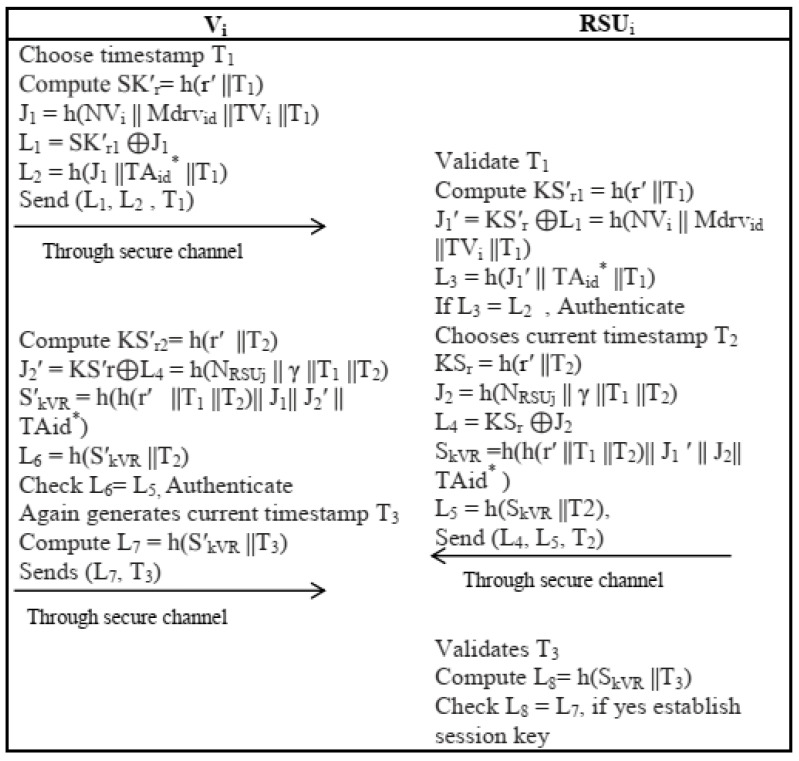
V2RSU Authentication and key establishment phase.

**Figure 6 sensors-22-04019-f006:**
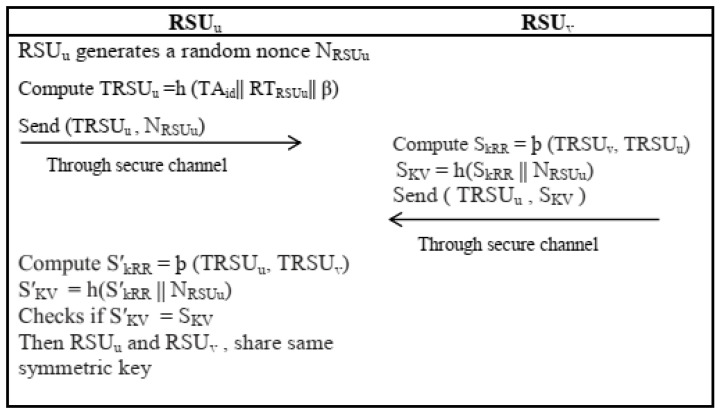
RSU2RSU Key Establishment Phase.

**Figure 7 sensors-22-04019-f007:**
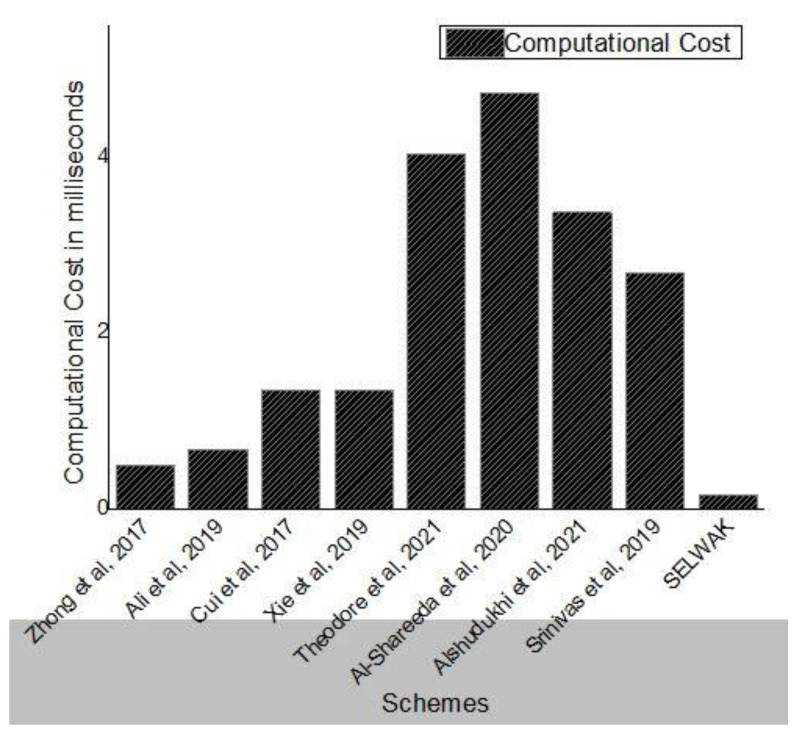
Computation Cost Comparison.

**Figure 8 sensors-22-04019-f008:**
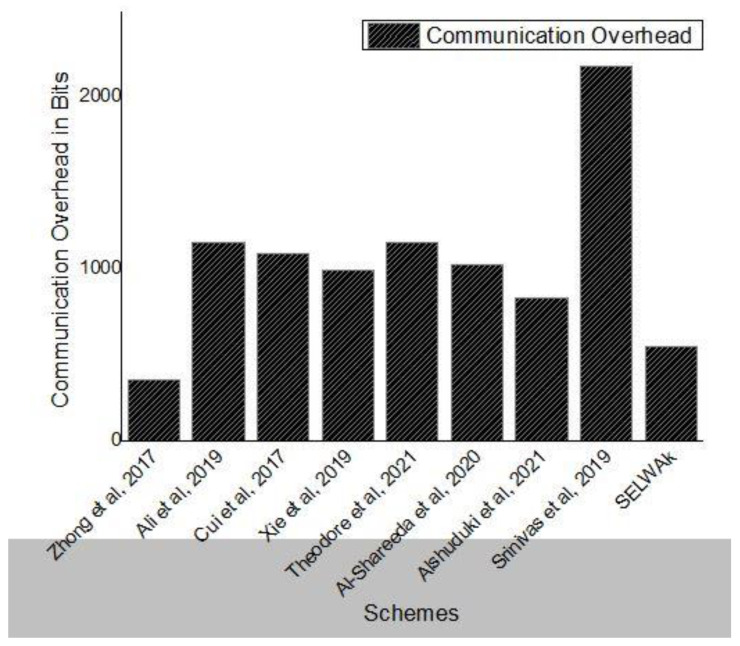
Communication Overhead Comparison.

**Table 1 sensors-22-04019-t001:** Notations used in the paper.

Notation	Description
$RSU_{j}$	*j*th Roadside Units
$V_{i}$	*i*th Vehicle
$Drv_{i}$	$\mathrm{Driver}\;\mathrm{of}\;\mathrm{the}\;\mathrm{vehicle}\;V_{i}$
$drv_{id}$	Identity of the driver
$RSUID_{j}$	$\mathrm{Identity}\;\mathrm{of}\;RSU_{j}$
$Mdrv_{id}$	Masked Identity of drivers
$TMRSU_{j}$	$\mathrm{Time}\;\mathrm{dependent}\;\mathrm{masked}\;\mathrm{identity}\;\mathrm{of}\;RSU_{j}$
$OBU_{i}$	*i*th Onboard Unit
$TA_{id}$	Identity of TA
α, *β*	160 bits secret keys of TA
$PWD_{i}$	Password chosen by drivers
$RT_{vi}$	$\mathrm{Registration}\;\mathrm{time}\;\mathrm{stamp}\;\mathrm{of}\;V_{i}$
$RT_{RSUj}$	$\mathrm{Registration}\;\mathrm{time}\;\mathrm{stamp}\;\mathrm{of}\;RSU_{j}$
*T*	Current time stamp
*N*	Random Nonce
Δ*T*	Max transmission delay
*h*(.)	One way hash function
||	Concatenation
⊕	Bitwise XOR operation

**Table 2 sensors-22-04019-t002:** Computation Cost Comparison.

Scheme	Total Computational Overhead	Total Execution Time (ms)
[17]	500*T*_h_	≈0.5
[19]	1*T*_pm_ − ECC + 1*T*_pa_ − ECC	≈0.6749
[20]	2*T*_pm_ − ECC + 1*T*_pa_ − ECC	≈1.3467
[21]	2*T*_pm_ − ECC + 1*T*_pa_ − ECC + *T*_h_	≈1.3477
[32]	6 *T*_pm_ − ECC + 1 *T*_pa_ − ECC + 4 *T*_h_	≈4.0348
[24]	7 *T*_pm_ − ECC + 2 *T*_pa_ − ECC + 4 *T*_h_	≈4.7128
[27]	5 *T*_pm_ − ECC + 1 *T*_pa_ − ECC + 4 *T*_h_	≈3.3661
[38]	4*T*_pm_ − ECC + 12*T*_h_	≈2.6992
SELWAK	16 *T*_h_ + 11 *T*_XOR_	≈0.016

**Table 3 sensors-22-04019-t003:** Communication Cost Comparison.

Schemes	Communication Overhead (Bits)
[17]	352 bits
[19]	1152 bits
[20]	1084 bits
[21]	992 bits
[32]	1152 bits
[24]	1024 bits
[27]	832bits
[38]	2176 bits
SELWAK	544 bits

## Data Availability

Not applicable.

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
