# Peer review of "SELWAK: A Secure and Efficient Lightweight and Anonymous Authentication and Key Establishment Scheme for IoT Based Vehicular Ad hoc Networks"

_sensors, 2022, doi:10.3390/s22114019_

Round 1
Reviewer 1 Report
In this paper, authors propose an efficient and secure lightweight anonymous mutual authentication and key establishment (SELWAK) for IoT based VANETs. Paper is clearly written, logically organized, very comprehensive and impressive amount of research and study has been carried out. The content is technically sound and contains sufficient interest.
The reviewer has some concerns.
- In the abstract, the performance analysis of this work must be described briefly.
- The paper lacks in-depth discussions in section 6.
- Some sentences are too long to make readers confused, and there are also some typos and grammar errors in this paper.
- The quality of the figures should be improved.
- Future works as an integral part should be included in the conclusions.
Author Response
Reviewer#1
Authors are thankful to the respected Reviewer-1 for his/her time and valuable comments to improve the quality of paper. All the changes suggested and recommended by the reviewer have been accommodated in the paper. Following is the detail of the reviewer concerns to the author and the authors’ response.
Reviewer#1, General Comments: In this paper, authors propose an efficient and secure lightweight anonymous mutual authentication and key establishment (SELWAK) for IoT based VANETs. Paper is clearly written, logically organized, very comprehensive and impressive amount of research and study has been carried out. The content is technically sound and contains sufficient interest.The reviewer has some concerns.
Author response: We are thankful to the reviewer for considering our article. To improve the quality of the paper, we have modified the paper according to the valuable comments of the reviewer. We believe that all the flaws were addressed in the current version.
Author action: To improve the quality of the paper, all the flaws were carefully corrected according to the valuable comments of the reviewer, the detail of which is given in the following responses.
- Reviewer#1, Concern # 1: In the abstract, the performance analysis of this work must be described briefly.
Author response: The performance analysis of proposed work is described in abstract
Author action: We updated the manuscript by describing the performance analysis of proposed work in the abstract section.
- Reviewer#1, Concern # 2: The paper lacks in-depth discussions in section 6.
Author response: The notations Tpm-ECC, Tpa-ECC and Th used in Table.2 represents Elliptic Curve Cryptographic points multiplication, Elliptic Curve Cryptographic points addition and one-way hash function respectively. As bitwise XOR operations take negligible time, therefore we have not considered it for performance evaluation.
We have considered the values 0.6718 ms, 0.0031 ms and 0.001 ms for various cryptographic operations like Tpm-ECC, Tpa-ECC and Th from existing experimental values[5,19, 27]. Computational cost of our scheme and existing schemes are compared in Table. 2. The schemes to which we compare our work include those of Zhong et al. [17], Ali et al. [19], Cui et al. [20] , Xie et al. [21], Li et al. [24], Al-shareeda et al.[27] and Jalawai et al. [32]. Zhong et al [17] proposed elliptic curve cryptography based conditional privacy preserving authentication scheme using registration list. The author used a registration list instead of a revocation list to overcome the communication overhead. An Elliptic curve cryptography based and Identity-Based Signature with Conditional Privacy-Preserving Authentication scheme and general one-way hash functions for V2V communication is proposed in [19]. Cui et al.[20]proposed a secure privacy-preserving authentication scheme for VANETs. This scheme uses ECC and identity based signatures for both V2I and V2V communication. Authors use binary search method and cuckoo filter method to improve success rate in batch signature verification. Xie et al. [21]proposed a robust and secure conditional privacy preserving scheme using identity-based authentication. The reliability and integrity of the messages is ensured using identity-based signatures for vehicle-to-vehicle communication and vehicle-to-infrastructure communication. Performance analysis shows that this scheme has high computational cost and communication overhead. To ensure secure communication in VANET, an authentication scheme based on ECC which satisfied privacy preservation is proposed in [27]. The total computational cost for SELWAK is 16Th+11TXOR , which is less than that of all compared schemes. Author action: We have elaborated some necessary discussion in section 6, which is highlighted.
Reviewer#1, Concern # 3: Some sentences are too long to make readers confused, and there are also some typos and grammar errors in this paper.
Author response: We have thoroughly read the paper and made necessary corrected where needed.
Author action: The corrections are highlighted in different section.
Reviewer#1, Concern # 4: The quality of the figures should be improved.
Author response: We have improved the quality of figures.
Author action: The quality of Figures 1, 2, 3, 4, 5, 6 have improved.
Reviewer#1, Concern # 5: Future works as an integral part should be included in the conclusions.
Author response: In the future work, the anonymous mutual authentication will be carry out using BAN Logic and some simulation platforms such as NS2, SUMO and OMNET++ to simulate VANETs.
Author action: We have included future works in Section 7 (Conclusion part).
Reviewer 2 Report
In this paper, the authors proposed a novel lightweight authentication and key establishment scheme for VANETs. Some suggestions are presented as follows.
- The motivation of the proposed scheme should be explained more before presenting the scheme, which may convince reader and help them to understand the key differences between the proposed work and existing solutions.
- In Figure 2, the authors proposed several steps, which are suggested to explain why we need them and what is difference between these steps and some published works.
- The simulation parts are suggested to provide more not only for computational and communication costs. Moreover, the mentioned published schemes are suggested to give brief introduction before comparison.
Author Response
Reviewer#2
Authors are thankful to the respected Reviewer-2 for his/her time and valuable comments to improve the quality of paper. All the changes suggested and recommended by the reviewer have been accommodated in the paper. Following is the detail of the reviewer concerns to the author and the authors’ response.
Reviewer#2, General Comments: Comments and Suggestions for Authors. In this paper, the authors proposed a novel lightweight authentication and key establishment scheme for VANETs. Some suggestions are presented as follows.
Author response: We are thankful to the reviewer for considering our article. To improve the quality of the paper, we have modified the paper according to the valuable comments of the reviewer. We believe that all the flaws were addressed in the current version.
Author action: To improve the quality of the paper, all the flaws were carefully corrected according to the valuable comments of the reviewer, the detail of which is given in the following responses.
Reviewer#2, Concern # 1: The motivation of the proposed scheme should be explained more before presenting the scheme, which may convince reader and help them to understand the key differences between the proposed work and existing solutions.
Author response:
In VANETs, the vehicles travel with high speeds and therefore schemes mentioned in literature are not optimal for such an environment. The OBU fixed in the vehicle has limited storage capacity, power supply and computational power. Various major cryptographic operations slow down the signing and verification process of traffic related messages. For example, Elliptic curve point multiplication and point addition are considered to be most time consuming operations in ECC based schemes. Therefore, it is difficult for RSU to verify a large number of vehicles in a short time period which are traveling with high speed in its communication range. It creates a high load on verifying entities that demand a secure and efficient lightweight and anonymous authentication and key establishment scheme for IoT based vehicular ad hoc networks.
Author action: We have explained motivations of proposed scheme in section 2, Sub section 2.1.
Reviewer#2, Concern # 2: In Figure 2, the authors proposed several steps, which are suggested to explain why we need them and what is difference between these steps and some published works.
Author response: In figure 2 involved five phases and provide lightweight security infrastructure for vehicular communication but some existing scheme uses more complex phases and increase the computational cost and communication overhead.
Author action: Phases are shown in figure 2.
Reviewer#2, Concern # 3: The simulation parts are suggested to provide more not only for computational and communication costs.
Author response: The notations Tpm-ECC, Tpa-ECC and Th used in Table.2 represents Elliptic Curve Cryptographic points multiplication, Elliptic Curve Cryptographic points addition and one-way hash function respectively. As bitwise XOR operations take negligible time, therefore we have not considered it for performance evaluation.
We have considered the values 0.6718 ms, 0.0031 ms and 0.001 ms for various cryptographic operations like Tpm-ECC, Tpa-ECC and Th from existing experimental values[5,19, 27]. Computational cost of our scheme and existing schemes are compared in Table. 2. The schemes to which we compare our work include those of Zhong et al. [17], Ali et al. [19], Cui et al. [20] , Xie et al. [21], Li et al. [24], Al-shareeda et al.[27] and Jalawai et al. [32]. Zhong et al [17] proposed an elliptic curve cryptography based conditional privacy preserving authentication scheme using registration list. The author used registration list instead of revocation list to overcome the communication overhead. An Elliptic curve cryptography based and Identity-Based Signature with Conditional Privacy-Preserving Authentication scheme and general one-way hash functions for V2V communication is proposed in [19]. Cui et al.[20]proposed a secure privacy-preserving authentication scheme for VANETs. This scheme uses ECC and identity based signatures for both V2I and V2V communication. Authors use binary search method and cuckoo filter method to improve success rate in batch signature verification. Xie et al. [21]proposed a robust and secure conditional privacy preserving scheme using identity-based authentication. The reliability and integrity of the messages is ensured using identity-based signatures for vehicle-to-vehicle communication and vehicle-to-infrastructure communication. Performance analysis shows that this scheme has high computational cost and communication overhead. To ensure secure communication in VANET, an authentication scheme based on ECC which satisfied privacy preservation is proposed in [27]. The total computational cost for SELWAK is 16Th+11TXOR , which is less than that of all compared schemes.
Author action: We have elaborated some necessary discussion in section 6, which is highlighted.
Reviewer#2, Concern # 4: Moreover, the mentioned published schemes are suggested to give brief introduction before comparison.
Author response: Zhong et al [17] proposed elliptic curve cryptography based conditional privacy preserving authentication scheme using registration list. The author used registration list instead of revocation list to overcome the communication overhead. An Elliptic curve cryptography based and Identity-Based Signature with Conditional Privacy-Preserving Authentication scheme and general one-way hash functions for V2V communication is proposed in [19]. Cui et al.[20]proposed a secure privacy-preserving authentication scheme for VANETs. This scheme uses ECC and identity based signatures for both V2I and V2V communication. Authors use binary search method and cuckoo filter method to improve success rate in batch signature verification. Xie et al. [21]proposed a robust and secure conditional privacy preserving scheme using identity-based authentication. The reliability and integrity of the messages is ensured using identity-based signatures for vehicle-to-vehicle communication and vehicle-to-infrastructure communication. Performance analysis shows that this scheme has high computational cost and communication overhead. To ensure secure communication in VANET, an authentication scheme based on ECC which satisfied privacy preservation is proposed in [27].
Author action: The brief introduction of published mentioned schemes is given in section 6, which is highlighted.
Reviewer 3 Report
I recommend to read the entire paper again because you have some mistakes in writing.
The paper presents very well the proposed schema, but maybe the authors can improve the presentation of the schema and the methods used to obtain the results.
For the results also, the paper can be improved by presenting clear the obtained results. In this form the reader should interpret the results from the presented analysis.
The conclusions section can be improved by adding some future plans of using this scheme.
Author Response
Reviewer#3
Authors are thankful to the respected Reviewer-3 for his/her time and valuable comments to improve the quality of paper. All the changes suggested and recommended by the reviewer have been accommodated in the paper. Following is the detail of the reviewer concerns to the author and the authors’ response.
Reviewer#3, Concern # 1:
Author response: I recommend reading the entire paper again because you have some mistakes in writing.
Author action: We have read the entire paper line by line and made corrections where needed.
Reviewer#3, Concern # 2: The paper presents very well the proposed schema, but maybe the authors can improve the presentation of the schema and the methods used to obtain the results. For the results also, the paper can be improved by presenting clear the obtained results. In this form the reader should interpret the results from the presented analysis
Author response
The notations Tpm-ECC, Tpa-ECC and Th used in Table.2 represents Elliptic Curve Cryptographic points multiplication, Elliptic Curve Cryptographic points addition and one-way hash function respectively. As bitwise XOR operations take negligible time, therefore we have not considered it for performance evaluation.
We have considered the values 0.6718 ms, 0.0031 ms and 0.001 ms for various cryptographic operations like Tpm-ECC, Tpa-ECC and Th from existing expermental values[5,19, 27].
Author action: The presentation of schemes and method to improve the result is shown in section 6.
Reviewer#3, Concern # 3: The conclusions section can be improved by adding some future plans of using this scheme .
Author response: In the future work, the anonymous mutual authentication will be carry out using BAN Logic and some simulation platforms such as NS2, SUMO and OMNET++ to simulate VANETs.
Author action: We have included future works in Section 7 (Conclusion part).